# U-CAN: Unsupervised Point Cloud Denoising with Consistency-Aware Noise2Noise Matching

**Junsheng Zhou**[1][*]    **Xingyu Shi**[1][*]    **Haichuan Song**[2][†]    **Yi Fang**[3]
**Yu-Shen Liu**[1][†]    **Zhizhong Han**[4]

School of Software, Tsinghua University, Beijing, China [1]
Computer Science and Technology, East China Normal University, Shanghai, China [2]
Center for AI and Robotics (CAIR), NYU Abu Dhabi, UAE [3]
Department of Computer Science, Wayne State University, Detroit, USA[4]

{zhou-js24,shixy22}@mails.tsinghua.edu.cn    hcsong@cs.ecnu.edu.cn
yfang@nyu.edu    liuyushen@tsinghua.edu.cn    h312h@wayne.edu

## Abstract

Point clouds captured by scanning sensors are often perturbed by noise, which have a highly negative impact on downstream tasks (e.g. surface reconstruction and shape understanding). Previous works mostly focus on training neural networks with noisy-clean point cloud pairs for learning denoising priors, which requires extensively manual efforts. In this work, we introduce **U-CAN**, an **U**nsupervised framework for point cloud denoising with **C**onsistency-**A**ware **N**oise2Noise matching. Specifically, we leverage a neural network to infer a multi-step denoising path for each point of a shape or scene with a noise to noise matching scheme. We achieve this by a novel loss which enables statistical reasoning on multiple noisy point cloud observations. We further introduce a novel constraint on the denoised geometry consistency for learning consistency-aware denoising patterns. We justify that the proposed constraint is a general term which is not limited to 3D domain and can also contribute to the area of 2D image denoising. Our evaluations under the widely used benchmarks in point cloud denoising, upsampling and image denoising show significant improvement over the state-of-the-art unsupervised methods, where U-CAN also produces comparable results with the supervised methods. Project page: https://gloriasze.github.io/U-CAN/.

## 1 Introduction

3D point clouds have been a fundamental representation in 3D computer vision and play a key role in autonomous driving [13], augmented/virtual reality [56] and robotics [10]. While in these real world applications, the point clouds captured with scanning sensors (e.g. LiDAR) contain unavoidable noise, which leads to large errors in 3D perception and understanding. Recent learning-based approaches [27, 26, 7] have shown convincing results in denoising point clouds with neural networks by learning denoising patterns with noisy-clean point cloud pairs. However, they are limited in the amount of clean 3D geometries which require manual efforts of human 3D CAD modelling.

A straightforward observation is that despite the limited clean models, the amount of real-captured noisy point clouds is growing rapidly everyday with the LiDARs in self-driving cars or consumer level digital devices in our daily life, such as iPhone. Consequently, it is desirable to learn denoising patterns by solely using the noisy data itself. The subsequent approaches, such as TotalDenoising [12], therefore turn to explore unsupervised point cloud denoising by leveraging a spatial prior term

---

[*]Equal contribution. [†]Corresponding authors.

39th Conference on Neural Information Processing Systems (NeurIPS 2025).

for total-level denoising. However, the current unsupervised approaches still struggle to predict precise clean point cloud while keeping high-fidelity local geometries due to the lack of sufficient constraints at local-level.

To solve these issues, we introduce U-CAN, an unsupervised learning framework for point cloud denoising with consistency-aware noise to noise matching. Instead of predicting a total-level denoising, we leverage a neural network to infer a multi-step denoising path for each point of a shape or scene with a point-wise noise to noise matching scheme. Specifically, we learn a mapping from one noisy point cloud to another with a novel loss function which enables a point-to-point matching for investigating denoising patterns from only noisy point clouds. The key idea of this noise to noise matching is to leverage the statistical reasoning to reveal the clean structures upon its several noisy observations.

Another challenge in predicting robust denoising arises from the unknown location of true surfaces when only noisy observations are available. This ambiguity can lead to unstable convergence due to inconsistencies in denoising results across different noisy observations. In response to this challenge, we introduce a novel consistency-aware constraint that specifically targets the denoising geometric consistency. We achieve this by minimizing the geometric differences from the denoising prediction of one noisy observation to the prediction of another. Furthermore, we demonstrate that the proposed consistency-aware denoising constraint is not limited to the 3D domain, which can also significantly contribute to the field of 2D image denoising. Extensive experiments demonstrate that the proposed U-CAN outperforms state-of-the-art methods in unsupervised point cloud denoising, upsampling and image denoising, where U-CAN even achieves comparable performances with the supervised methods. Our main contributions can be summarized as follows.

- We introduce U-CAN, a novel framework for unsupervised point cloud denoising by designing a neural network to infer a multi-step denoising path for each point of a noisy observation with a novel noise to noise matching loss.

- We propose a general constraint on the denoising geometric consistency across different denoising predictions, which significantly improves the denoising performance in both 3D and 2D domain.

- We achieve state-of-the-art results in unsupervised denoising for both point clouds and images under widely used benchmarks, while also delivering performance comparable to supervised methods. U-CAN is also capable of unsupervised point upsampling through denoising.

## 2 Related Work

### 2.1 Traditional Point Cloud Denoising

Point cloud denoising plays a key role in robust 3D understanding, as the point clouds captured by scanners often contain unavoidable noise. Traditional methods for optimizing noisy point clouds include local surface fitting [1, 14], sparse representation [3, 44], and graph filtering [41, 55], all of which use geometric priors for denoising. Local surface fitting methods, such as the widely used moving least squares (MLS) method [2] and its robust extensions [35, 9], approximate the point cloud with a smooth surface using simple function approximators and project the points onto this newly formed surface for denoising. Other techniques, like jet fitting [5] and the parameterization-free local projector operator (LOP) [25, 14], have also been developed for point cloud denoising. Sparsity-based methods [3, 44] address denoising by initially predicting surface normals through optimization problems with sparse constraints. Graph-based methods [41, 55] represent point clouds using graphs and use graph filters for denoising. The graph-based methods are sensitive to the noise distributions due to the potential instability in graph construction.

### 2.2 Learning-based Point Cloud Denoising

The deep learning based approaches for 3D point cloud [46, 67, 74, 54, 49, 18, 64, 16, 73, 75, 57, 62] have largely advanced point cloud processing tasks, such as upsampling [51, 24], surface reconstruction [71, 72, 4, 42, 29, 65, 11, 70, 15, 69, 34, 61], consolidation [33, 52], normal estimation [23, 63], generation [76, 68, 77, 48, 66] and denoising [40, 36, 27, 26, 19, 31, 58, 47, 60]. With the emergence

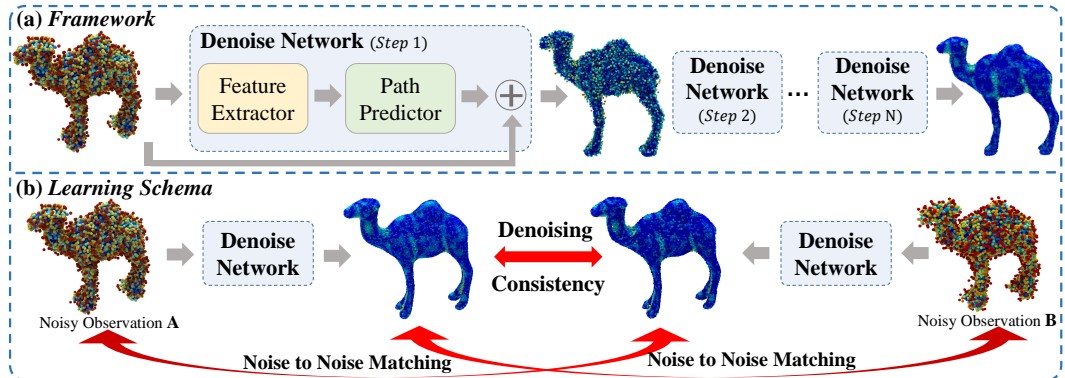

Figure 1: Overview of our method. (a) We design a multi-step denoising framework to gradually filter the noisy point cloud. (b) We introduce a novel learning schema for unsupervised learning of point cloud denoising by proposing two constraints, i.e., Noise to Noise Matching loss and Denoising Consistency loss.

of neural networks for point cloud processing, such as PointNet [37], PointNet++ [38] and DGCNN [46], learning representations on point sets for denoising has achieved convincing performances. PointCleanNet [40] is the pioneer of learning-based point denoising which introduces a neural network based on the PointNet model [37, 38]. PointFilter [59] advances PointCleanNet with a special designed filter modeling. The following work GPDNet [36] introduces graph convolutional networks for improving and stabilizing denoising process, introducing the strong capability of graph networks in handling complex geometric data to the point cloud denoising task. DMRDenoise [26] introduces a new perspective for denoising by adopting an innovative downsample-upsample framework. More recently, ScoreDenoise [27] attempts to estimate the gradient field around each point and iteratively update the position of each point. IterativeFPN [7] simulates a real iterative filtering process internally to reduce noise.

## 2.3 Unsupervised Point Cloud Denoising

Previous learning-based approaches merely focus on learning denoising patterns with noisy-clean point cloud pairs and are limited in the amount of clean 3D shapes which require manual efforts of human CAD modeling. TotalDenoising [12] is the most relevant work of ours which explores unsupervised point denoising by leveraging a spatial prior term for total-level denoising. However, it struggles to predict precise predictions with high-fidelity local geometries. The reason is that TotalDenoising only involves the global constraint and lacks the local-level constraint which plays the key role in producing detailed predictions. DMRDenoise [26] and ScoreDenoise [27] also provide an unsupervised version by introducing the total-denoising loss, but both of them face the same problem as TotalDenoising [12]. A recent work [28] introduces an unsupervised approach to over-fit each noisy point cloud for learning signed distance functions, where each point cloud takes about more than 10 minutes to converge. We focus on the learning-based point cloud denoising which enables a fast inference.

Different from these works, we learn a data-driven matching from one noisy point cloud to another with a novel loss function which enables point-to-point matching at local-level. This brings high-fidelity denoising results. We further introduce denoising consistency constraint for consistency-aware predictions with improved accuracy.

## 3 Architecture of U-CAN

**Problem Statement.** We design a neural network with a novel learning schema for unsupervised point cloud denoising. Current methods train neural networks to denoise a point cloud by matching it with its paired clean point cloud. Different from these supervised methods, we do not require any clean point clouds as supervision, and learn to filter a noisy observation $\mathcal{P}_a$ of a 3D shape or scene $\mathcal{S}$ with only other several noisy observation $\mathcal{P}_b$ of $\mathcal{S}$.

**Overview.** The overview of proposed U-CAN is shown in Fig. 1. We will start from our denoise network in Sec. 3.1 and introduce the noise to noise mapping schema with a novel point-wise matching loss in Sec. 3.2. We then present a novel constraint on denoising consistency in Sec. 3.3 and transfer it to enhance the unsupervised image denoising task in Sec. 3.4.

## 3.1 Denoise Network

Given a noisy point cloud $\mathcal{P}_a$ as input, we design a multi-step denoising framework to gradually filter $\mathcal{P}_a$ for achieving a cleaned point cloud $\mathbb{C}_a$. As illustrated in Fig. 1 (a), the *Denoise Network* at each step consists of a *Feature Extractor* and a *Path Predictor*. We implement the *Feature Extractor* as a series of dynamic EdgeConv from DGCNN [46] with residual connections for achieving robust representations, while the *Path Predictor* is composed with several linear layers to predict the moving path for each point from the extracted features.

During the $i$-th step of the denoising process, the Denoise Network $f_i$ takes the filtered point clouds $\mathcal{C}_a^{i-1}$ from the previous step (the noisy point cloud $\mathcal{P}_a$ for the initial step) as input. It then predicts a distinct moving path $\triangle p_i$ for pulling each point to attain the filtered point cloud $\mathcal{C}_a^i$ at the current step as $\mathcal{C}_a^i = \mathcal{C}_a^{i-1} + \triangle p_i$. The final prediction $\mathbb{C}_a$ is obtained by moving $\mathcal{P}_a$ gradually, formulated as:

$$\mathbb{C}_a = \mathcal{P} + f_1(\mathcal{P}_a) + \sum_{i=2}^{N} f_i(\mathcal{C}_a^{i-1}), \tag{1}$$

where $N > 1$ is the number of steps.

## 3.2 Noise to Noise Matching

The common practice for predicting the clean point cloud from its noisy observations is to train a neural network with Noise to Clean Matching supervisions to minimize the distance between the predicted $\mathbb{C}_a$ with the ground truth clean point cloud $G_a$, formulated as:

$$\mathcal{L}_{N2C} = \mathcal{D}(\mathbb{C}_a, G_a), \tag{2}$$

where $\mathcal{D}(\cdot, \cdot)$ is a distance metric, typically the Chamfer Distance.

**Preview Noise2Noise.** Previously, Noise2Noise [22] has been proposed for unsupervised image denoising by encouraging a denoised image to resemble other noisy observations of the same image. Given the appealing results in 2D domain, it seems that we can denoise 3D point cloud unsupervisedly by simply transferring the success of Noise2Noise into 3D domain. However, the conclusion of Noise2Noise is built upon the one-to-one matching correspondences, as the pixels in images. The correspondences support the key assumption of Noise2Noise that the noisy values at the same pixel location of different observations are random realizations of a distribution around a clean pixel value. While the point clouds are irregular and unordered with no correspondences where a naive reproduction of Noise2Noise do not work for 3D point clouds.

A balancing approach for adapting unsupervised denoising in point clouds is to design a total-level loss for global denoising without specifying the correspondences like TotalDenoising [12], yet it struggles to predict precise clean point cloud while keeping local geometries with only the coarse constraint. NoiseMap [28] introduces an over-fitting approach to learn signed distance functions for each noisy observation with the Noise2Noise mechanism in local-level, but fails in generalizing to new observations.

**One-to-One Point Correspondences.** As discussed above, we justify that the key factor preventing the adaption of Noise2Noise schema in 3D domain is the lack of 3D correspondences. To solve this issue, we aim to build an one-to-one correspondence with a specific matching for each point. Instead of manually defining the point correspondences, we explore a suitable distance metric $\mathcal{D}$ that potentially contains the one-to-one point correspondences as the optimizing target, which satisfies the assumptions for Noise2Noise and can naturally enable the unsupervised denoising for 3D point clouds. In practice, we use Earth Moving Distance (EMD) as a suitable implementation of $\mathcal{D}$. The EMD between two point clouds $X$ and $Y$ is formulated as:

$$D_{\text{EMD}}(X, Y) = \min_{\phi: X \to Y} \sum_{x \in X} ||x - \phi(x)||_2, \tag{3}$$

where $\phi$ is a one-to-one correspondence. With the Earth Moving Distance which potentially contains the point correspondences as the distance metric, we successfully adopt the Noise2Noise schema for unsupervised point cloud denoising. Specifically, as shown in Fig. 1 (b), given two noisy observations $\mathcal{P}_a$ and $\mathcal{P}_b$ randomly selected from a set of noisy observations at each epoch, we train the *Denoise Network* with the Noise to Noise Matching loss to push the denoised point cloud $\mathbb{C}_a$ to be similar to another noisy point cloud $\mathcal{P}_b$, and vice versa for $\mathbb{C}_b$. The Noise to Noise Matching loss is formulated as:

$$\mathcal{L}_{N2N} = \mathcal{D}_{\text{EMD}}(\mathbb{C}_a, \mathcal{P}_b) + \mathcal{D}_{\text{EMD}}(\mathbb{C}_b, \mathcal{P}_a). \tag{4}$$

With this loss, U-CAN leverages the statistical reasoning among multiple noisy observations and effectively infers clean structures.

## 3.3 Denoising Consistency Constraint

Another issue that none of the previous unsupervised denoising works on 2D or 3D domains noticed is that the unsupervised noise to noise matching schema struggles to produce a consistent denoising prediction with different noisy observations as input. This leads to ambiguous optimizations for the detailed geometries. For the supervised approaches, this is not a problem since each noisy input has a distinct clean point cloud as the target. While in the situation of unsupervised denoising, there is no true surface locations provided, and only multiple noisy observations are available as inputs and targets, which makes it hard for the neural networks to learn consistent predictions.

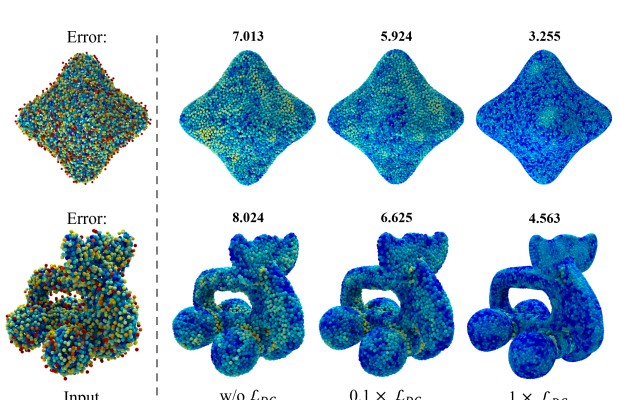

Figure 2: Illustrations on the effect of proposed constraint on denoising consistency. The noise errors indicate the Chamfer distance between the denoised and the clean point clouds.

Driven by this observation, we propose a novel constraint on denoising consistency for learning consistency-aware denoising patterns. Specifically, we push the denoised prediction of one noisy observation to be consistent with the denoised prediction of another noisy observation with a special designed loss, formulated as:

$$\mathcal{L}_{\text{DC}} = \mathcal{D}_{\text{EMD}}(\mathbb{C}_a, \mathbb{C}_b), \tag{5}$$

where $\mathbb{C}_a$ and $\mathbb{C}_b$ are the denoised predictions achieved by Eq. (1), respectively. With the simple but effective term, U-CAN is able to produce more consistent-aware predictions and further improve the denoising results at detailed local geometries.

We provide an illustration as shown in Fig. 2 to show the advantage of our proposed constraint $\mathcal{L}_{\text{DC}}$. We train U-CAN for learning point cloud denoising without $\mathcal{L}_{\text{DC}}$ and show the result as "w/o $\mathcal{L}_{\text{DC}}$". We than visualize the denoising predictions of U-CAN trained with low coefficient (i.e. $0.1\times$) and high coefficient (i.e. $1\times$) of $\mathcal{L}_{\text{DC}}$, shown as "$0.1\times \mathcal{L}_{\text{DC}}$" and "$1\times \mathcal{L}_{\text{DC}}$". The comparisons demonstrate the effectiveness of $\mathcal{L}_{\text{DC}}$ where better performances are achieved with larger coefficient of $\mathcal{L}_{\text{DC}}$.

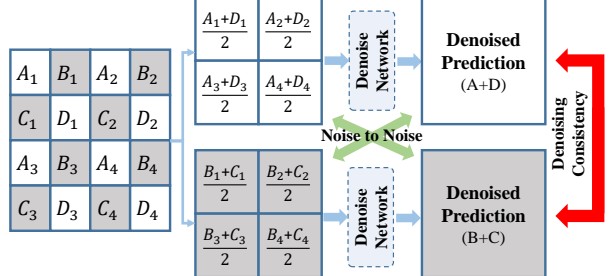

Figure 3: Transferring the denoising consistency constraint of U-CAN to the unsupervised image denoising.

Table 1: Denoising comparisons under PUNet dataset. CD$\times10^4$ and P2M $\times10^4$.The best results under the unsupervised (Un-Sup) point cloud denoising setting are highlighted.

| | Point Number | 10K(Sparse) | | | | | | 50K(Dense) | | | | | |
|---|---|---|---|---|---|---|---|---|---|---|---|---|---|
| | Noise | 1% | | 2% | | 3% | | 1% | | 2% | | 3% | |
| | Model | CD | P2M | CD | P2M | CD | P2M | CD | P2M | CD | P2M | CD | P2M |
| Classic | Bilateral [8] | 3.646 | 1.342 | 5.007 | 2.018 | 6.998 | 3.557 | 0.877 | 0.234 | 2.376 | 1.389 | 6.304 | 4.730 |
| | Jet [5] | 2.712 | 0.613 | 4.155 | 1.347 | 6.262 | 2.921 | 0.851 | 0.207 | 2.432 | 1.403 | 5.788 | 4.267 |
| | MRPCA [32] | 2.972 | 0.922 | 3.728 | 1.117 | 5.009 | 1.963 | 0.669 | 0.099 | 2.008 | 1.003 | 5.775 | 4.081 |
| | GLR [55] | 2.959 | 1.052 | 3.773 | 1.306 | 4.909 | 2.114 | 0.696 | 0.161 | 1.587 | 0.830 | 3.839 | 2.707 |
| Supervised | PCNet [40] | 3.515 | 1.148 | 7.469 | 3.965 | 13.067 | 8.737 | 1.049 | 0.346 | 1.447 | 0.608 | 2.289 | 1.285 |
| | GPDNet [36] | 3.780 | 1.337 | 8.007 | 4.426 | 13.482 | 9.114 | 1.913 | 1.037 | 5.021 | 3.736 | 9.705 | 7.998 |
| | DMR [26] | 4.482 | 1.722 | 4.982 | 2.115 | 5.892 | 2.846 | 1.162 | 0.469 | 1.566 | 0.800 | 2.632 | 1.528 |
| | ScoreDenoise [27] | 2.521 | 0.463 | 3.686 | 1.074 | 4.708 | 1.942 | 0.716 | 0.150 | 1.288 | 0.566 | 1.928 | 1.041 |
| Un-Sup | TTD [12] | 3.390 | **0.826** | 7.251 | 3.485 | 13.385 | 8.740 | 1.024 | **0.314** | 2.722 | 1.567 | 7.474 | 5.729 |
| | DMR-TTD | 7.897 | 5.026 | 9.257 | 6.119 | 10.946 | 7.569 | 2.137 | 1.567 | 3.223 | 2.498 | 5.572 | 4.669 |
| | ScoreDenoise-TTD | 3.107 | 0.888 | 4.675 | 1.829 | 7.225 | 3.726 | 0.918 | 0.265 | 2.439 | 1.411 | 5.303 | 3.841 |
| | **Ours** | **2.497** | 1.105 | **3.234** | **1.255** | **3.666** | **1.842** | **0.835** | 0.609 | **0.975** | **0.675** | 2.479 | 1.863 |

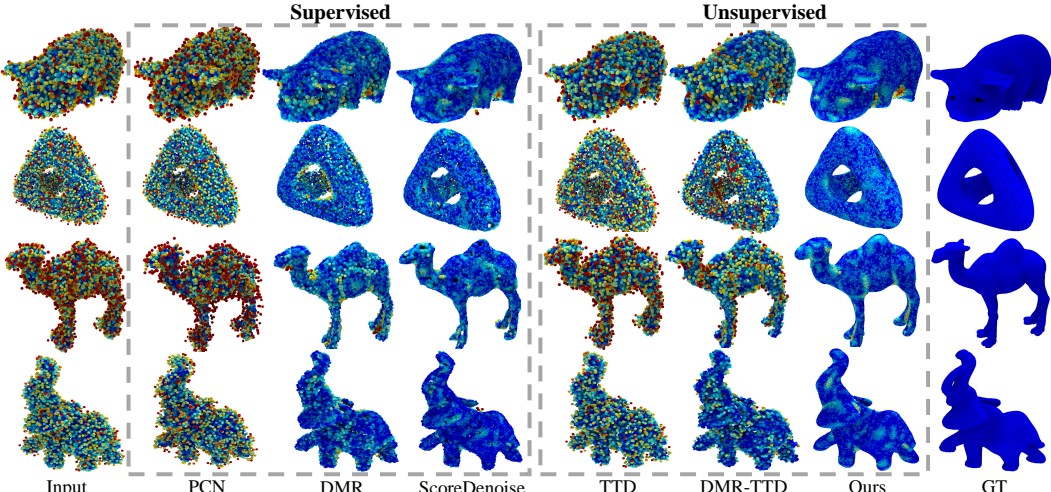

Figure 4: Visual comparisons under PUNet dataset. The noise errors at each point is shown in color, where the points closer to the ground truth surface are represented with bluer color, indicating lower error. And those with higher error are represented with redder color.

## 3.4   Transferring U-CAN to Image Denoising

We further justify that the observation in Sec. 3.3 is not limited in the unsupervised point cloud denoising, but is a common issue that also occurs in the unsupervised image denoising task. Therefore, we believe the proposed constraint for denoising consistency in Eq. (5) can also contribute to the area of 2D image denoising. We demonstrate the effectiveness of $\mathcal{L}_{\text{DC}}$ by adapting it to the state-of-the-art work ZS-N2N [30] on unsupervised image denoise.

The overview of modified ZS-N2N is shown in Fig. 3. ZS-N2N separates an image into two downsampled sub-images and treats them as two noisy observations to learn a residual-based noise to noise matching for image denoising. We further introduce our proposed constraint on denoising consistency to ZS-N2N by minimizing the differences between one denoised sub-image and the other denoised sub-image.

## 4   Experiments

### 4.1   Point Cloud Denoising on Synthetic Data

**Dataset and Metrics.** For the experiments on synthetic shapes, we follow ScoreDenoise [27] to train our network on the PUNet [53] dataset. We split the dataset into training and testing sets with the same setting as ScoreDenoise [27]. Poisson disk sampling algorithm is used to sample point clouds from

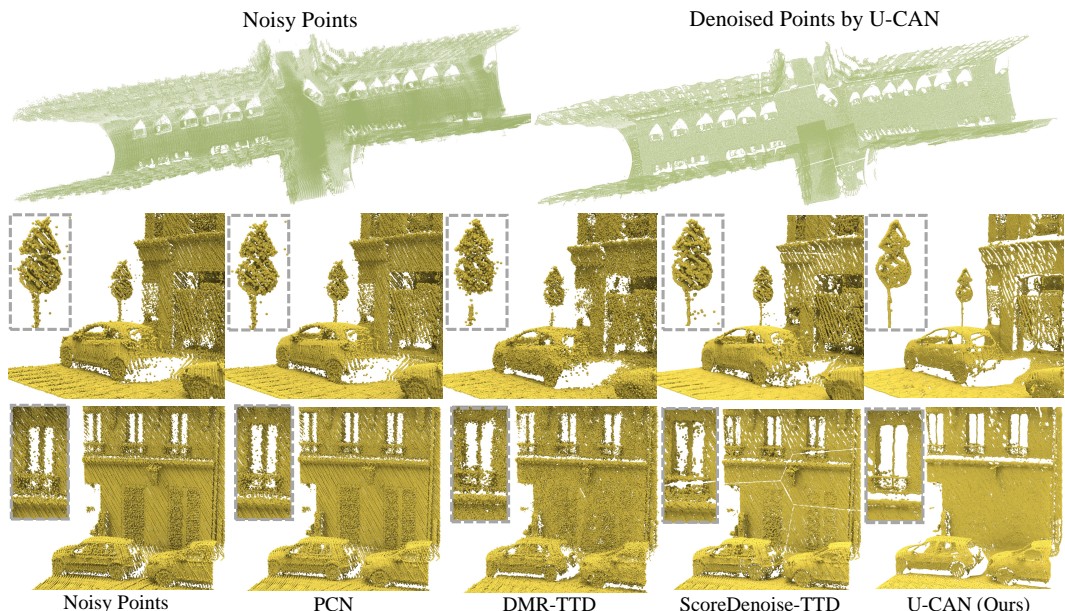

Figure 5: Denoising on the real scans under Paris-rue-Madame dataset. **Top:** The visualization of the noisy points and denoised points obtained by U-CAN under the whole scene. **Bottom**: The visual comparison with the supervised and unsupervised approaches on the local scene geometries.

the meshes at two resolutions: 10k and 50k points and the Gaussian noise is subsequently introduced at three different levels of standard deviations, i.e., 1%, 2% and 3% of the bounding sphere's radius. Following previous works PCNet [40] and DMR [26], we split point clouds into patches before being fed into the model, where the patch size is set to 1K. We evaluate the performance of U-CAN and other baselines under the commonly used metrics L2 Chamfer distance (CD) and the point-to-mesh distance (P2M), following previous methods [27, 26]

**Comparisons.** We quantitatively compare the proposed U-CAN with the state-of-the-art methods for both supervised and unsupervised point cloud denoising in Tab. 1. This includes classic optimization-based methods such as Bilateral [8], Jet [5], MRPCA [32], GLR [55]; supervised learning-based methods like PCNet [40], GPDNet [36], DMR [26], ScoreDenoise [27], and PointFilter [59]; and unsupervised methods including TTD [12], as well as unsupervised adaptations of DMR [26] and ScoreDenoise [27] with the TTD loss, shown as 'DMR-TTD' and 'ScoreDenoise-TTD'.

The comparative analysis of methods using synthetic data is presented in Tab. 1 and illustrated in Fig. 4. Traditional optimization-based point cloud denoising methods rely heavily on geometric priors to inform their smoothing algorithms and show increased sensitivity to noises with unseen variances, leading to degradation in denoising performance. For unsupervised denoising, the TTD [12] fails to produce high-fidelity local geometries with only the global constraints. The unsupervised versions of DMR [26] and ScoreDenoise [27] which leverage the same constraint as TTD, share same limitations of TTD and presents sub-optimal performance at both low and high resolutions due to the lack of local-level constraints.

As presented, our model significantly outperforms previous unsupervised denoising methods, especially for noises with large variances, and can even rival the results of supervised methods in the majority of cases. In particular, at the 10K resolution and under noise levels of 2% and 3%, our method outperforms all other supervised and unsupervised methods in the evaluation.

We provide the visual comparison among the state-of-the-art supervised and unsupervised point cloud denoising methods in Fig. 4. The error at each point in the point cloud is depicted in color. Points that are closer to the ground truth surface are shown in blue, indicating lower error, while those with higher error are shown in red. As shown in the figure, our results produces significantly more visual-appealing denoising results compared to other unsupervised approaches and even some supervised ones.

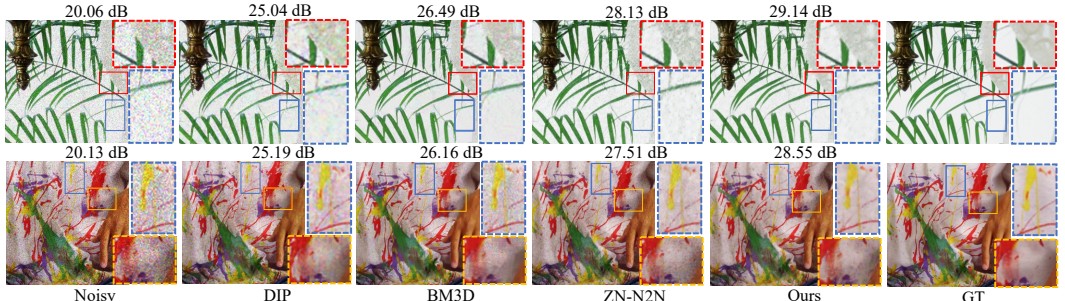

Figure 6: Visual comparison of unsupervised image denoising under McMaster18 dataset.

## 4.2 Point Cloud Denoising on Scanned Data

For demonstrating the capability of U-CAN to handle real-world point cloud noises, we conduct evaluations under the Paris-rue-Madame dataset [43] which is obtained from real world using laser scanners. We directly leverage the U-CAN model trained on PUNet dataset for evaluating, without requiring extra training. The visualization of denoised scene point cloud is shown in Fig. 5. Since the ground truth point cloud is not available, our evaluations are primarily qualitative, focusing on visual assessments rather than quantitative metrics.

As shown in Fig. 5, U-CAN preserves the intricate details better and yields a cleaner and smoother surface. On the scene in the top row, our method demonstrates a marked improvement over the other compared methods, particularly around complex structures like trees and cars. In the bottom row, we observe that windows are denoised with greater clarity and cleanliness with the proposed U-CAN. We also produce accurate denoising results of the surrounding structures such as the walls and vehicles.

## 4.3 Evaluatioins in Image Denoising

**Dataset and Metrics.** We further evaluate the proposed denoising consistency constraint for improving the image denoising qualities. For evaluating in the image denoising task, we follow ZS-N2N [30] to conduct experiments under the McMaster18 dataset [20]. Our evaluation setting keeps the same as ZS-N2N, and center-crop the images into patches of size $256 \times 256$. We examine under poisson noise with noise levels $\lambda = 10, 25, 50$. We leverage the commonly-used PSNR in dB as the evaluation metric.

**Comparisons.** We compare the proposed image denoising adaption of U-CAN with the state-of-the-art methods for unsupervised image denoising, including the dataset-based Noise2Clean (N2C), Neighbour2Neighbour (NB2NB) [17], Noise2Void (N2V) [21], and the dataset free methods BM3D [6], DIP [45], Self2Self (S2S) [39] and ZS-N2N [30]. We show the quantitative comparison in Tab. 2, where the denoising consistency constraint demonstrates superior performance compared to the previous methods. Specifically, by introducing the proposed denoising consistency constraint into ZS-N2N, we achieve significant improvements of nearly 1 dB over the baseline ZS-N2N. The visual comparison is shown in Fig. 6. The denoised images

Table 2: Unsupervised image denoising under McMaster18 dataset. The PSNR scores in dB are reported. Best results are marked in bold and the second-best results are underlined.

| Noise | | Method | $\lambda$ known? | $\lambda = 50$ | $\lambda = 25$ | $\lambda = 10$ |
|---|---|---|---|---|---|---|
| Poisson | dataset-based | N2C | yes | 29.89 | 28.20 | 26.42 |
| | | | no | 28.62 | 27.51 | 24.32 |
| | | NB2NB | yes | 29.41 | 27.79 | 25.95 |
| | | | no | 28.03 | 27.66 | 24.58 |
| | | N2V | yes | 27.86 | 25.65 | 23.47 |
| | | | no | 26.34 | 25.52 | 22.07 |
| | dataset-free | BM3D | no | 27.33 | 24.77 | 21.59 |
| | | DIP | - | 28.73 | 27.37 | 24.67 |
| | | S2S | - | 27.55 | 27.24 | 26.39 |
| | | ZS-N2N | - | 30.36 | 28.41 | 25.75 |
| | | Ours | - | **31.03** | **29.14** | **26.52** |

of U-CAN are more accurate and with more details than previous state-of-the-art methods. This is particularly clear in areas of high-frequency information, such as edges, textures, and intricate patterns, where our method maintains the integrity of these details while effectively reducing noise.

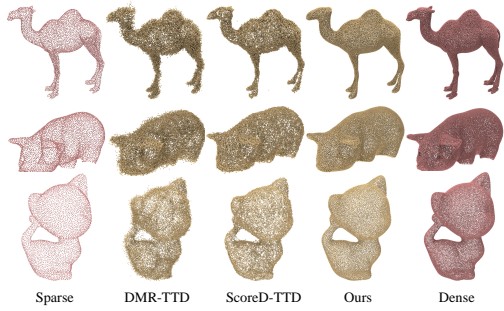

Figure 7: Visual Comparison under PU-Net.

Sparse    DMR-TTD    ScoreD-TTD    Ours    Dense

| #Points | 5K | | 10K | |
|---|---|---|---|---|
| | CD | P2M | CD | P2M |
| PU-Net [53] | 3.445 | 1.669 | 2.862 | 1.166 |
| ScoreDenoise [27] | 1.696 | **0.295** | 1.454 | **0.181** |
| DMR-TTD [26, 12] | 5.846 | 4.045 | 4.710 | 3.833 |
| ScoreD-TTD [27, 12] | 3.286 | 1.889 | 2.403 | 1.683 |
| Ours | **1.532** | 0.585 | **1.212** | 0.587 |

Table 3: Point cloud upsampling results under PU-Net dataset.

## 4.4 Point Upsampling via Denoising

**Implementation.** We further justify that the proposed U-CAN is applicable in point cloud upsampling task, without requiring sparse-dense point cloud pairs and even without requiring the clean point clouds. We follow ScoreDenoise [27] to conduct experiments in the point cloud upsampling task. Specifically, given a sparse point cloud with $M$ points as the input, we add Gaussian noise to it for $r$ times independently, resulting in a noisy dense point cloud containing $rM$ points. We then feed the merged noisy point cloud to the trained U-CAN model to get the final upsampled point cloud by predicting the denoised points.

**Dataset and Metrics.** We follow ScoreDenoise [27] to conduct the point cloud upsampling experiments under the PU-Net dataset. We report the evaluation metrics of chamfer distance (CD) and point-to-mesh (P2M).

**Comparisons.** We compare our proposed U-CAN with the classical updsampling network PU-Net [53]. We further apply the adaption from denoising to upsampling to the state-of-the-art unsupervised point cloud denoising methods DMR-TTD and ScoreDenoise-TTD and report their upsampling performances. The quantitative results are shown in Tab. 3, where our method significantly outperforms DMR-TTD and ScoreDenoise-TTD, and also achieve better performance than the supervised method PU-Net designed for the upsampling task. Note that U-CAN does not require (1) sparse-to-dense point cloud pairs and (2) clean point clouds, where the only required data is the noise point clouds themselves. We further provide the visual comparison of point cloud upsampling with previous state-of-the-art unsupervised denoising methods in Fig. 7

## 4.5 Ablation Studies

**Noise-to-Noise Matching Loss $\mathcal{L}_{N2N}$.** We investigate the role of EMD-based one-to-one point correspondences in $\mathcal{L}_{N2N}$. As shown in Tab. 4, replacing EMD with CD leads to suboptimal patterns, while using Density-aware Chamfer Distance (DCD) [50] causes severe divergence. These results highlight the necessity of one-to-one matching for effective unsupervised denoising.

**Denoising Consistency Loss $\mathcal{L}_{DC}$.** To justify the effectiveness of constraint $\mathcal{L}_{DC}$, we remove it and vary the underlying distance metric. Without $\mathcal{L}DC$, performance significantly drops (e.g., CD

Table 4: Ablation studies on the framework and loss designs.

| Dataset: PU | | 10K, 1% | | 10K, 2% | | 10K, 3% | |
|---|---|---|---|---|---|---|---|
| $\mathcal{L}_{N2N}$ | $\mathcal{L}_{DC}$ | CD | P2M | CD | P2M | CD | P2M |
| CD | EMD | 15.48 | 11.36 | 17.42 | 13.22 | 21.39 | 17.01 |
| DCD | EMD | broken | - | - | - | - | - |
| **EMD** | **EMD** | **2.497** | **1.105** | **3.234** | **1.255** | **3.666** | **1.842** |
| EMD | ✗ | 2.208 | 0.725 | 3.731 | 1.631 | 7.218 | 4.468 |
| EMD | CD | 2.108 | 0.650 | 3.717 | 1.633 | 7.230 | 4.469 |
| EMD | DCD | **2.036** | **0.608** | 3.358 | 1.367 | 6.847 | 4.144 |
| **EMD** | **EMD** | 2.497 | 1.105 | **3.234** | **1.255** | **3.666** | **1.842** |

Table 5: Ablation studies on step numbers in the denoise network.

| Dataset: PU | 10K, 1% | | 10K, 2% | | 10K, 3% | |
|---|---|---|---|---|---|---|
| Ablation | CD | P2M | CD | P2M | CD | P2M |
| 1 step | 2.676 | 1.046 | 3.903 | 1.700 | 5.251 | 2.720 |
| 2 steps | 2.606 | 1.159 | 3.507 | 1.670 | 4.096 | 2.069 |
| 3 steps | 2.492 | 1.096 | 3.246 | 1.554 | 3.704 | 1.878 |
| **4 steps** | 2.497 | 1.105 | 3.234 | 1.255 | **3.666** | **1.842** |
| 5 steps | 2.514 | 1.118 | 3.388 | **1.151** | 3.746 | 1.903 |
| 6 steps | 2.509 | 1.107 | **3.225** | 1.235 | 3.753 | 1.857 |
| 7 steps | **2.470** | **1.080** | 3.321 | 1.243 | 3.785 | 1.866 |

increases from 3.66 to 7.22 under '10K, 3%'), indicating its critical role in enforcing consistent predictions across noisy inputs. EMD again proves to be the most effective metric.

**Number of Denoising Steps.** We study the impact of varying the number of denoising steps $N$ from 1 to 7. As shown in Tab. 5, performance improves up to $N{=}4$, beyond which gains saturate or slightly degrade. Thus, 4 steps offer a good trade-off between accuracy and efficiency.

## 5   Conclusion

In this work, we introduce **U-CAN**, an **U**nsupervised framework for point cloud denoising with **C**onsistency-**A**ware **N**oise2Noise matching. We train a neural network to infer a denoising path for each point of a shape with a noise to noise matching scheme. Our novel loss enables statistical reasoning on noisy point cloud observations. We also introduce a novel constraint on the denoising geometry consistency for learning consistency-aware denoising patterns. Our evaluation for point cloud denoising and image denoising demonstrates that even without clean supervision, U-CAN also produces comparable denoising results with the state-of-the-art supervised methods.

## 6   Acknowledgement

This work was supported by Deep Earth Probe and Mineral Resources Exploration – National Science and Technology Major Project (2024ZD1003405), and the National Natural Science Foundation of China (62272263), and in part by Kuaishou. Junsheng Zhou is also partially funded by Baidu Scholarship.

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
