# OpenReview forum: "U-CAN: Unsupervised Point Cloud Denoising with Consistency-Aware Noise2Noise Matching"
_NeurIPS.cc/2025/Conference — NeurIPS 2025 poster_

### Official Review · Reviewer_hTqa · 2025-06-11

**Clarity:** 3
**Significance:** 3
**Originality:** 2
**Rating:** 4
**Confidence:** 3

**Summary:**

The paper introduces  U-CAN , an unsupervised framework for point cloud denoising that leverages a  Consistency-Aware Noise2Noise Matching  approach. U-CAN employs a neural network to infer multi-step denoising paths for noisy point clouds, using a novel  Noise to Noise Matching loss  to learn denoising patterns without requiring clean ground truth data. Additionally, it introduces a  Denoising Consistency loss  to ensure geometric consistency across multiple noisy observations, enhancing robustness. The framework is evaluated on synthetic (PUNet) and real-world (Paris-rue-Madame) datasets, achieving state-of-the-art performance in unsupervised point cloud denoising and comparable results to supervised methods. U-CAN also demonstrates applicability to 2D image denoising (McMaster18 dataset) and unsupervised point cloud upsampling, showcasing its versatility.

**Questions:**

See weakness.

**Ethical Concerns:**

["NO or VERY MINOR ethics concerns only"]

**Limitations:**

The authors adequately address limitations. They note the qualitative nature of evaluations on real-world data (Paris-rue-Madame) due to the lack of ground truth  and acknowledge challenges in predicting high-fidelity local geometries compared to supervised methods. The paper appropriately marks societal impact questions as “NA” (not provided in the excerpt but assumed per NeurIPS guidelines), as the work is foundational and does not involve high-risk applications. The transparency about limitations is commendable, and no critical points appear to be missing.

**Paper Formatting Concerns:**

No.

**Quality:**

3

**Strengths And Weaknesses:**

Strengths:

1. The paper introduces U-CAN, an unsupervised framework that leverages consistency-aware Noise2Noise matching to denoise point clouds without clean supervision. By using a neural network to infer multi-step denoising paths and a novel loss function for statistical reasoning on noisy observations, U-CAN overcomes the need for manually annotated clean point clouds, addressing a key challenge in prior supervised methods . The framework's design, which adapts Noise2Noise to 3D point clouds via Earth Mover's Distance (EMD), effectively handles the unordered nature of point sets.

2. The proposed Denoising Consistency Constraint (L_DC) ensures geometric consistency across denoised predictions from different noisy inputs, mitigating ambiguity in unsupervised learning. This constraint not only improves local geometry preservation in point clouds but also generalizes to 2D image denoising, demonstrating its versatility . The combination of L_N2N (Noise to Noise Matching loss) and L_DC enables U-CAN to produce high-fidelity results while maintaining computational efficiency.

3. U-CAN outperforms state-of-the-art unsupervised methods in point cloud denoising, upsampling, and image denoising.

Weaknesses :

1. Unverified Effectiveness with Alternative Backbone Networks
The study focuses on a specific denoising network architecture but does not explore how the proposed Consistency-Aware Noise2Noise losses (Lₙ₂ₙ and L_DC) perform when integrated with other backbone networks (e.g., PointNet++,  or transformer-based architectures). Verifying their generality across different network designs is crucial to confirm the framework’s adaptability and robustness.

2.Lack of Evaluation on Indoor Point Cloud Datasets
The paper does not assess U-CAN on indoor point cloud benchmarks such as ScanNet, which are critical for real-world applications in robotics, AR/VR, and autonomous interior navigation. Indoor scenes often contain complex structures (e.g., furniture, partitions) and varying noise distributions (e.g., sensor reflections or occlusions), posing unique challenges for denoising.

---

> ### Author Rebuttal · Authors · 2025-07-31
>
> We sincerely appreciate Reviewer hTqa for the acknowledgment of our work and constructive feedback. We respond to each question below.
>
> **Q1: Alternative backbone networks.**
>
> We thank reviewer hTqa for the insightful suggestion regarding the integration of the proposed framework with alternative backbone networks. Following this advice, we replaced the EdgeConv-based denoising network with PointNet++ and PointTransformer to evaluate the generality of U-CAN across different architectures. The results are reported in Table F below.
>
> ***Table F: Effectiveness of U-CAN with different backbones.***
>
> | Dataset: PU               |          | 10K, 1%   |         | 10K, 2%   |         | 10K, 3%   |         |
> |---------------------------|----------|----------|---------|----------|---------|----------|---------|
> | Ablation                  |          | CD       | P2M     | CD       | P2M     | CD       | P2M     |
> | U-CAN (PointNet++)        |          | 3.427    | 1.803   | 4.393    | 2.424   | 5.047    | 2.969   |
> | U-CAN (PointTransformer)  |          | 2.547    | 1.204   | 3.277    | 1.639   | 3.684    | 1.892   |
> | **Ours**                  |          | **2.497**| **1.105**| **3.234**| **1.255**| **3.666**| **1.842** |
>
> As demonstrated, the U-CAN framework consistently achieves robust denoising performance across different backbone networks. PointNet++ is a relatively older backbone that provides stable but suboptimal results, while PointTransformer, a more modern architecture, achieves performance comparable to our default denoising model. Notably, we directly integrate these backbones into U-CAN without the need for manual tuning of settings or hyperparameters specific to each architecture. These results highlight the strong generality, adaptability, and robustness of the proposed framework.
>
> **Q2: Evaluation on indoor point cloud datasets like ScanNet.**
>
> We sincerely thank the reviewer hTqa for the valuable advice and fully recognize the importance of evaluating U-CAN under real-world indoor scene scans. To provide a more comprehensive evaluation of U-CAN's performance, we conducted an experiment on the ScanNet dataset, a real-world indoor dataset consisting of noisy point clouds obtained from scanning sensors. These point clouds contain complex structures (e.g., furniture, partitions) and exhibit varying noise distributions (e.g., sensor reflections and occlusions). We apply U-CAN to predict denoised point clouds from these scans and report the numerical metrics comparing the denoised predictions with the ground truth clean meshes provided in the ScanNet dataset. The results, along with comparisons to state-of-the-art unsupervised point denoising methods, are presented in Table B below, where U-CAN achieves the best performance.
>
>
> ***Table B. Comparisions under indoor ScanNet dataset.***
>
> | Method        | CD | P2M |
> |---------------|----------|----------|
> | DMR-TTD   | 2.629    |   2.381  |
> | ScoreDenoise-TTD  | 2.474     | 2.524     |
> | **Ours**      | **1.798**     | **1.897**     |
>
> U-CAN achieves superior performance on both the indoor and outdoor scene point clouds scanned in the real world, demonstrating its remarkable capability in real-world downstream applications in robotics, autonomous driving, AR/VR, and autonomous interior navigation.

---

### Official Review · Reviewer_y6U3 · 2025-06-14

**Clarity:** 3
**Significance:** 3
**Originality:** 2
**Rating:** 4
**Confidence:** 5

**Summary:**

This paper proposes an unsupervised multi-step point cloud denoising network, which leverages a noise-to-noise matching loss and a geometric consistency loss to guide training. The proposed approach achieves competitive performance in unsupervised denoising tasks.

**Questions:**

Please refer to the "weakness" section mentioned above.

**Ethical Concerns:**

["NO or VERY MINOR ethics concerns only"]

**Final Justification:**

The authors have addressed most of my concerns, and I am satisfied with the additional experiments, clarifications, and overall presentation. My main reservation remains the similarity to Noise2NoiseMapping in the underlying principle, though I acknowledge the differences in learning scheme and framework, as well as the novelty of enforcing cross-scan consistency. The claimed extension to other modalities still requires stronger validation.
Overall, while the work may not be highly groundbreaking, it contains merits and acknowledged novelty. I can raise my score to 4.

**Limitations:**

yes

**Quality:**

3

**Strengths And Weaknesses:**

**Strengths**:
- The paper is well-organized and easy to follow.
- The topic of unsupervised point cloud denoising is meaningful.
- Enforcing consistency among denoised results from different scans of the same object/scene is a reasonable and insightful strategy.

**Weaknesses**: \
Major Concerns:
- 1. The motivation and core framework of this paper closely resemble the *Noise2NoiseMapping* approach proposed in [1]. Several key statements are similar with the phrasing of [1]. For example:\
line 39 of this paper: "**The key idea** of this noise to noise matching is to leverage the **statistical reasoning** to reveal the clean structures upon its several noisy observations" \
line 140: "The correspondences support the key assumption of Noise2Noise that **the noisy values at the same pixel location of different observations are random realizations of a distribution around a clean pixel value.**" \
While sharing similar motivations with prior work can be acceptable, the proposed solution itself also exhibits notable similarities. Both this paper and [1] achieve unsupervised denoising by ensuring that the denoised output of a noisy scan aligns closely (in terms of EMD) with other scans of the same object/scene. Notably, the use of EMD as a loss metric was presented as a contribution in [1], and has since been adopted in other works such as [2].
- 2. The multi-step denoising design should not be claimed as a contribution (line 54), as similar multi-stage frameworks have been explored since [3]. Does the proposed multi-step denoising framework apply a loss function at each step for supervision? Prior studies have shown that applying supervision at each stage can enhance denoising performance.
- 3. The experimental section does not include a comparison with [1], which is a highly relevant baseline.
- 4. As shown in Table 1, the proposed method performs worse in terms of P2M under low noise levels compared to moderate or high noise. This is counterintuitive since low-noise data should be easier to denoise. Could the authors provide an explanation for this?

Minor Concerns:
- 1. The related work section misses several recent fully-supervised denoising methods proposed after [3]. Even though the paper focuses on unsupervised methods, a brief discussion of these approaches would provide better context.
- 2. Lines 110 and 116 use the word "schema," which appears to be a typo and should be corrected to "scheme."
- 3. Please clarify what the symbol P denotes in Equation (1).
- 4. In Figure 5, the proposed method appears to over-smooth some structural details, particularly on flat surfaces such as walls. This might be a common issue in unsupervised methods, and the authors have mentioned this limitation. Still, it would be helpful to give some possible solutions.

[1] Learning Signed Distance Functions from Noisy 3D Point Clouds via Noise to Noise Mapping\
[2] Denoising Point Clouds in Latent Space via Graph Convolution and Invertible Neural Network\
[3] IterativePFN: True Iterative Point Cloud Filtering

---

> ### Author Rebuttal · Authors · 2025-07-31
>
> We deeply appreciate the reviewer y6U3 for the thoughtful feedback and time invested in evaluating our work. We respond to each question below.
>
> **Q1: The difference on motivation and core framework with NoiseMap.**
>
> We would like to thank the reviewer y6U3 for pointing out the similarity in motivation between U-CAN and NoiseMap [1]. U-CAN did share a similar motivation with NoiseMap in transferring the success of the Noise2Noise scheme from the 2D image domain to 3D point cloud denoising. However, the core framework and methodology of U-CAN differ significantly from those of NoiseMap in four key aspects:
>
> **1. Learning scheme and framework.** NoiseMap is an overfitting approach which requires more than **10 minutes** to converge for each noisy observation and cannot generalize to new observations. In contrast, U-CAN is a standard learning-based point cloud denoising framework that learns data-driven patterns during training. The resulting model can directly generalize to unseen noisy observations with a fast inference time (**10 seconds**). As a result, U-CAN is suitable for downstream tasks, while NoiseMap’s reliance on per-instance overfitting hinders its use in real-world scenarios.
>
> **2. Difference in core contributions.** (a) A key innovation of U-CAN is the *Denoising Consistency Constraint*  which aims to learn consistent denoising patterns by minimizing geometric differences between the denoised outputs of different noisy observations. We are the first to identify and address an overlooked issue in prior unsupervised denoising methods for 2D and 3D domains: the Noise2Noise scheme often fails to produce consistent predictions from different noisy inputs, leading to ambiguity in optimizing fine geometric details.
>
> (b) Beyond 3D vision, another key contribution of U-CAN is its extension to unsupervised denoising in other modalities. The proposed *Denoising Consistency Constraint* is a general term applicable to both 3D point cloud and 2D image denoising. We introduce a modified version of SOTA unsupervised image denosing method ZS-N2N, achieving significant performance gains. We believe this general constraint can benefit unsupervised denoising across a broader range of modalities, including but not limited to 2D and 3D vision—for example, speech denoising.
>
> **3. Representation difference.** NoiseMap adopts Signed Distance Functions (SDFs) as its 3D shape representation, which are inherently limited to closed surfaces. In contrast, U-CAN predicts denoising paths directly on point clouds, enabling it to represent 3D shapes with arbitrary topologies. To evaluate the performance of both NoiseMap and U-CAN on non-closed 3D shapes, we conducted additional experiments on the DeepFashion3D dataset, which consists of 3D garments with open surfaces. Quantitative comparisons are presented in Table C below.
>
> ***Table C: Comparisons with NoiseMap under DeepFashion3D.***
>
> | Method   | CD     | P2M    |
> |------------|--------|--------|
> | NoiseMap | 6.392  | 3.173  |
> | **Ours**   | **1.036** | **0.586** |
>
> **4. Model architecture.** The architectures of NoiseMap and U-CAN differ significantly. NoiseMap is an overfitting method that uses a simple MLP as its neural network backbone. In contrast, U-CAN is designed as a robust learning-based point cloud denoising model with efficient inference, adopting a multi-step architecture built upon dynamic EdgeConv modules.
>
> **Q2: Lack of comparison with NoiseMap.**
>
> We would like to clarify that we have conducted comprehensive comparison between U-CAN and NoiseMap in terms of both performance and efficiency. Section C.1 of the Appendix is dedicated to this comparison: Table 9 reports quantitative results on CD, P2M, and runtime, while Figure 9 provides visual comparisons. These results demonstrate that U-CAN achieves performance comparable to NoiseMap, both quantitatively and qualitatively.
>
> For the reviewer’s convenience, we include Table 9 from the Appendix below.
>
> ***Table 9 of the Appendix: Comparisons with overfitting method NoiseMap.***
>
> | Method   | CD     | P2M    | Time     |
> |------------|--------|--------|----------|
> | NoiseMap [28] | 4.221  | 1.847  | 627.0 s  |
> | **Ours**   | **3.666** | **1.842** | **10.9 s** |
>
> The reason we do not include NoiseMap in the comparison in the main paper is that it is an overfitting approach, requiring more than 10 minutes to converge for each noisy observation and failing to generalize to new observations. With redundant overfitting, NoiseMap achieves good performance but cannot be used in real world applications. Therefore, we excluded NoiseMap from Table 1, as comparing it with other learning-based baselines would not be fair. Nonetheless, U-CAN achieves comparable performance to NoiseMap, with ***60x*** faster runtime, demonstrating its robustness and superiority.
>
>
> **Q3: Claim of multi-step denoising.**
>
> We fully acknowledge that the multi-step denoising design was first introduced in IterativePFN and should not be claimed as a contribution. We will revise the statement in L.54 to *"We introduce U-CAN, a novel framework for unsupervised point cloud denoising by **leveraging** a neural network to infer a multi-step denoising path ..."*
>
> **Q4: Loss function setting of multi-step denoising.**
>
> Indeed, your understanding is accurate: the multi-step denoising framework applies a loss function at each step for supervision. We observed the same phenomenon as previous studies, where applying loss functions at each step results in more robust point cloud denoising. The results of our ablation studies on the loss settings are reported in Table D below.
>
> ***Table D: Ablation studies on the multi-step loss setting.***
>
> | Dataset: PU    |   | 10K, 1%   |      | 10K, 2%   |      | 10K, 3%   |      |
> |---------------------|----------|----------|---------|----------|---------|----------|---------|
> | Ablation   |  | CD    | P2M     | CD       | P2M     | CD   | P2M     |
> | Only supervise last step    |     | 2.990    | 1.272   | 4.091    | 2.057   | 5.542 | 3.231   |
> | **Ours**   |     | **2.497** | **1.105** | **3.234** | **1.255** | **3.666** | **1.842** |
>
> **Q5: Performance under low noise levels.**
>
> We would like to clarify that the "worse" P2M performance at 1% noise is actually due to a version difference in the P2M implementation used in ScoreDenoise. For direct comparison with our most relevant baseline, ScoreDenoise, we directly report the numerical results from their paper. However, the P2M implementation in PyTorch3D used by ScoreDenoise was unstable at the time of their results, leading to abnormally good outcomes that cannot be reproduced with the latest version, particularly at lower noise levels (e.g., 1%). For more details, please refer to Issue 12 in the ScoreDenoise GitHub repository. Recent supervised methods, such as IterativePFN (CVPR 2023), DP-LTS (CVPR 2024), and 3DMambaIPF (AAAI 2025), have confirmed the issue and follow the stable PyTorch3D P2M function. With this stable function, P2M values are significantly higher. Please refer to Table 1 in IterativePFN for details.
>
> We justify that we use the same stable PyTorch3D implementation as IterativePFN/DP-LTS/3DMambaIPF, which also face the issue of larger P2M values. To ensure a fair comparison, we will update Table 1 in U-CAN with results from IterativePFN. We update the P2M results for ScoreDenoise-TTD using the stable implementation and report them in Table E below. Note that all P2M results in U-CAN, except for Table 1, are evaluated with the consistent stable PyTorch3D version and do not suffer from this issue.
>
> ***Table E: Updated P2M comparison with ScoreDenoise-TTD.***
>
> | Dataset: PU| | 10K | | | 50K | |
> |------------|--------|--------|--------|--------|--------|--------|
> |    | 1%   | 2%  |3%| 1%    | 2%   |3%|
> | ScoreDenoise-TTD | 1.355  | 2.185  |  3.765  | 0.655  | 1.720 | 4.166  |
> | **Ours**   | **1.105** | **1.255** | **1.842** | **0.609** | **0.675**| **1.863**|
>
> Furthermore, we would like to clarify that U-CAN consistently outperforms previous unsupervised methods across all noise levels, including those lower than 1% and higher than 3%. We have conducted comprehensive comparisons in Section C of the Appendix, where Table 7 and Table 8 demonstrate that U-CAN significantly outperforms the unsupervised SOTA methods GLR, DMR-TTD, and ScoreDenoise-TTD at noise levels of 0.2%, 0.5%, 4%, and 5%.
>
> **Q6: Missing related works.**
>
> We greatly thank reviewer y6U3 for the advice on improving the related work section. We follow the suggestions to include more discussions of latest full-supervised denoising methods. We will add the statements in Sec.2.2, after L.91:
>
> *"StraightPCF advances IterativePFN by learning to move noisy points to the clean surfaces along the shortest path. PD-LTS designs an invertible network for achieving noise disentanglement in the latent space. P2PBridge and SuperPC incorporate diffusion models into point cloud denoising. 3DMambaIPF  leverages the powerful Mamba model for more efficient point denoising."*
>
> The corresponding references will be added.
>
> **Q7: Over-smooth results on flat surfaces.**
>
> We acknowledge that over-smoothing on flat surfaces (e.g., walls) is a common issue in unsupervised methods. We thank reviewer y6U3 for the insightful comment. A practical solution is to reduce the number of denoising steps, especially in scenes with rich structural details. In our observations, a small number of steps (e.g., 1 to 2) is sufficient to remove strong noise like artifacts near windows or road signs, while also preserving flat surface details. We will include visualizations on the Paris-rue-Madame dataset using the 1 to 2 step version of U-CAN in the revised submission, since figures cannot be uploaded during the rebuttal phase of NeurIPS.
>
> **Q8: Typos and clarifications.**
>
> The symbol $P$ in Equation (1) is actually the noisy point cloud $P_a$ for the initial step. We will correct the symbol and other typographical mistakes in our revision.

---

> > ### Comment · Reviewer_y6U3 · 2025-08-05
> >
> > Thank you for the detailed rebuttal. The authors have successfully addressed most of my concerns. I am satisfied with the additional experiments and the provided clarifications, as well as the overall organization and writing of the manuscript.
> >
> > My primary reservation continues to be the similarity between the proposed method and Noise2NoiseMapping. I maintain my view that the fundamental principle of both works is similar: they achieve unsupervised denoising by ensuring that the denoised output of a noisy scan aligns closely with other scans of the same object/scene. However, I accept the authors' argument regarding the differences in the learning scheme and framework. I also appreciate the novelty of enforcing consistency across the denoised results from different scans. Regarding the claimed extension to unsupervised denoising in other modalities, I believe this claim is not sufficiently substantiated without more thorough validation.
> >
> > Overall, while I still argue that the paper may not be exceptionally insightful, as a more groundbreaking approach might have introduced a fundamentally different principle for unsupervised point cloud denoising, the work possesses merits and acknowledged novelty. Moreover, the extra evaluation and clarification are worth serious consideration. I intend to raise my score to 4.

---

> > > ### Author Response · Authors · 2025-08-05
> > >
> > > Dear Reviewer y6U3,
> > >
> > > Many thanks for the positive assessment and for acknowledging the efforts we made in the rebuttal. We truly appreciate your recognition of the additional experiments, clarifications, and the overall writing and organization of our manuscript.
> > >
> > > We also understand and respect your continued concerns regarding the similarity in underlying principles with prior work. Your feedback has provided us with valuable insights, and we are glad that our clarification on the differences in learning schemes and the proposed denoising consistency framework was helpful.
> > >
> > > Regarding your point on the extension to denoising consistency constraint in other modalities, we acknowledge that further validation is indeed necessary. Following your suggestion, we plan to further explore the application of consistency constraints for unsupervised denoising in other modalities, in order to validate and generalize the applicability of our approach beyond point clouds and images.
> > >
> > > Thank you once again for your constructive feedback and for your willingness to raise the score. We sincerely appreciate your thoughtful engagement with our work.
> > >
> > > Best regards,
> > >
> > > The authors

---

> ### Author Response · Authors · 2025-08-08
>
> Dear Reviewer y6U3,
>
>
> We sincerely appreciate your willingness to raise the score and your constructive feedback, which has substantially improved our work. It appears the **final rating** has not yet been updated in the system. At your convenience, would you kindly submit it?
>
> Thank you again for positive assessment and thoughtful engagement with our work.
>
> Best regards,
>
> The authors

---

> > ### Author Response · Authors · 2025-08-09
> >
> > Dear Reviewer y6U3,
> >
> > We truly appreciate your willingness to raise our score. Your constructive feedback has been invaluable in improving our work, and we have greatly enjoyed the insightful discussions with you.  As the rebuttal period is coming to a close, we kindly ask if you could submit the updated score in the system at a time that is convenient for you.
> >
> > Thank you again for your positive assessment and  time invested in evaluating our work.
> >
> >
> >
> > Best regards,
> >
> > The authors

---

### Official Review · Reviewer_25Zo · 2025-06-30

**Clarity:** 3
**Significance:** 3
**Originality:** 2
**Rating:** 3
**Confidence:** 4

**Summary:**

This paper extends Noise2Noise in the field of 2D image denoising to 3D data, and achieves high-quality point cloud denoising through the proposed "Noise to Noise Matching" and "Denoising Consistency Constraint".

**Questions:**

1. This experiment adds various noises to the clean PUNet dataset to synthesize network inputs. I wonder if the noise distributions of Pa and Pb are consistent. In other words, are Pa and Pb obtained by adding the same level of noise to the same clean shape/scene, but with different noise distributions?
2. If the noise distributions of Pa and Pb are different, I believe there is a risk of data leakage. Since Pa and Pb are contaminated with noise in different areas, the network could directly learn from the complementary regions of Pa and Pb without any denoising.
3. I also noticed that the authors added real-world noise to PUNet for their experiments. I would like to know how U-CAN performs in denoising single-frame point clouds that contain noise in the real world.
4 . This work contains two branches, Pa->Ca and Pb->Cb. I am curious whether the networks of these two branches are two independent networks or the same network with shared weights? Compared with existing unsupervised algorithms, does U-CAN have an advantage in model complexity?
5. U-CAN contains two parallel denoising branches. Does this greatly increase the inference time of the model? I hope the author can provide relevant data.

**Ethical Concerns:**

["NO or VERY MINOR ethics concerns only"]

**Final Justification:**

After checking the authors' response and other reviews. And I lean to keep the original rating.

**Limitations:**

The core idea of this work comes from Noise2Noise for 2D image denoising, but I didn't see any relevant content in the introduction of the manuscript.
The latest comparison algorithm in this work is from 2021, which makes it difficult to explain the superiority of U-CAN.

**Paper Formatting Concerns:**

Lacks analysis of model complexity, comparison with the latest algorithms. Unclear model parameters, and the motivation is not well-supported by relevant literature.

**Quality:**

3

**Strengths And Weaknesses:**

The structure is clear, and the proposed U-CAN can achieve data denoising across multiple modalities with good performance.

---

> ### Author Rebuttal · Authors · 2025-07-31
>
> We deeply appreciate the reviewer 25Zo for the thoughtful feedback and time invested in evaluating our work. We respond to each question below.
>
> **Q1: The noise distribution of different noisy observations.**
>
> The noisy observations $P_a$ and $P_b$ are corrupted with noise sampled from the same distribution. We clarify that the training samples can be obtained in two ways: a) multiple noisy observations of the same object. This is done by adding noise drawn from the same distribution multiple times to the same object. The introduced noises are just different samples from the same noise distribution. This setting is realistic and supported by modern LiDAR systems, which can capture 10–30 noisy observations per second with a consistent noise distribution. b) several subsets of a dense noisy observation. For example, we can achieve 5 training instances of 10k points from a noisy observation of 50k points.  Since all subsets originate from the same noisy observation, they share the same noise distribution.
>
> In conclusion, both approaches produce training data with a consistent noise distribution. An ablation study comparing these two input strategies is presented in Table 12 of the Appendix, where results demonstrate the effectiveness of both.
>
> For the reviewer’s convenience, we include Table 12 from the Appendix below.
>
> ***Table 12 of the Appendix: Ablation studies on input patterns.***
>
> | Dataset: PU         |          | 10K, 1%   |         | 10K, 2%   |         | 10K, 3%   |         |
> |---------------------|----------|----------|---------|----------|---------|----------|---------|
> | Ablation            |          | CD       | P2M     | CD       | P2M     | CD       | P2M     |
> | Multiple 𝒫          |          | **2.497** | **1.105** | **3.234** | **1.255** | 3.666 | **1.842** |
> | Subsets of One 𝒫    |          | 2.580    | 1.231   | 3.287    | 1.661   | **3.605** | 1.869   |
>
> **Q2: Can the network learn from complementary regions of $P_a$ and $P_b$?**
>
> We would like to clarify that all regions of the training noisy observations $P_a$ and $P_b$ are corrupted with noise from the same distribution and at the same level. There are no "complementary regions" that could result in data leakage. Actually, we add the noises to the entire global shape, where each single point at every location of the global shape is perturbed by noise sampled from the same distribution.
>
> **Q3: The performance of U-CAN under point clouds contain noise in the real world.**
>
> We fully recognize the importance of evaluating U-CAN on real-world scans. In fact, we have conducted experiments and demonstrated the performance of U-CAN on the Paris-rue-Madame dataset, which is an outdoor real-world dataset acquired using laser scanners and contains complex scanning noise. Please refer to Sec.4.2 of the main paper for more details on the experimental settings and visualizations. A comprehensive comparison between U-CAN and both supervised and unsupervised baselines, including PCN, DMR-TTD, and ScoreDenoise-TTD, is presented in Figure 5 of the main paper.
>
> Paris-rue-Madame is a real-world **outdoor** scanned dataset. For a more comprehensive evaluation on the performance of U-CAN, we conduct another experiment on a real-world **indoor** scanned dataset ScanNet. The input data from ScanNet consists of noisy point clouds captured by scanning sensors. We apply U-CAN to denoise these scans and evaluate the results by comparing the predictions with the ground-truth clean meshes provided in the dataset. Quantitative results and comparisons with state-of-the-art unsupervised denoising methods are shown in Table B below, where U-CAN achieves the best performance.
>
> ***Table B. Comparisions under indoor ScanNet dataset .***
>
> | Method        | CD | P2M |
> |---------------|----------|----------|
> | DMR-TTD   | 2.629    |   2.381  |
> | ScoreDenoise-TTD  | 2.474     | 2.524     |
> | **Ours**      | **1.798**     | **1.897**     |
>
> U-CAN achieves superior performance on both the indoor and outdoor scene point clouds scanned in the real world, demonstrating its remarkable capability in real-world downstream applications in robotics, autonomous driving, AR/VR, and indoor navigation.
>
> **Q4: The model weights in two branches of U-CAN, model complexity and inference time.**
>
> The two branches in U-CAN are actually the same network with shared weights, and therefore do not introduce any additional model complexity. U-CAN contains *~ 789K* parameters, which is slightly more than ScoreDenoise (*~ 187K*) and DMR (*~ 225K*). However, we argue that U-CAN remains a lightweight network, as most recent (supervised) point cloud denoising methods contain over 1M parameters.
>
> We also clarify that U-CAN achieves comparable runtime to other unsupervised point cloud denoising methods despite having more parameters, thanks to its efficient EdgeConv-based architecture. Table A below presents the inference time comparison between U-CAN, state-of-the-art unsupervised baselines (ScoreDenoise-TTD, DMR-TTD), the supervised baseline PCN, and the traditional algorithm GLR.
>
> ***Table A. Runtime comparisons under PU-Net dataset.***
>
> | Method        | Time (s) |
> |---------------|----------|
> | GLR (Traditional)          | 101.20    |
> | PCN (Supervised)          | 167.75   |
> | DMR-TTD   | 11.08    |
> | ScoreDenoise-TTD  | 18.70     |
> | **Ours**      | **10.90**     |
>
>
> **Q5: The introduction of Noise2Noise.**
>
> We thank reviewer 25Zo for the valuable suggestion to include more descriptions of Noise2Noise in the introduction. In fact, we have devote a full paragraph (L.139-L.144) on Sec.3.2 to introducing the original work and core techniques of Noise2Noise. To provide a more comprehensive background, we will add more content of Noise2Noise to the introduction. Specifically, we will add the sentense before L.34:
>
> *"Noise2Noise [19] is a pioneering framework for unsupervised image denoising, which learns to produce clean images by encouraging the denoised output to resemble other noisy observations of the same image at pixel-level. Inspired by Noise2Noise [19], we introduce U-CAN ..."*
>
>
> **Q6: The compared algorithms.**
>
> We would like to clarify that most recent works focus on supervised point cloud denoising. In the area of learning-based unsupervised point cloud denoising, the compared methods DMR-TTD and ScoreDenoise-TTD are the most recent and relevant to U-CAN. Additionally, we have included a comparison with another recent overfitting-based approach, NoiseMap, published at ICML 2023, in Table 9 and Figure 9 of the Appendix. U-CAN demonstrates comparable performance to NoiseMap, both quantitatively and qualitatively.
>
> For the reviewer’s convenience, we include Table 9 from the Appendix below.
>
> ***Table 9 of the Appendix: Comparisons with over-fitting method NoiseMap.***
>
> | Method   | CD     | P2M    | Time     |
> |------------|--------|--------|----------|
> | NoiseMap [28] (ICML 2023) | 4.221  | 1.847  | 627.0 s  |
> | **Ours**   | **3.666** | **1.842** | **10.9 s** |
>
> The reason we do not include NoiseMap in the comparison in the main paper is that it is an overfitting approach, requiring more than 10 minutes to converge for each noisy observation and failing to generalize to new observations. With redundant overfitting, NoiseMap achieves good performance but cannot be used in real world applications. Therefore, we excluded NoiseMap from Table 1, as comparing it with other learning-based baselines would not be fair. Nonetheless, U-CAN achieves comparable performance to NoiseMap, with ***60x*** faster runtime, demonstrating its robustness and superiority.

---

> ### Author Response · Authors · 2025-08-06
>
> Dear Reviewer 25Zo,
>
> As the reviewer-author discussion period is nearing its end, we are sincerely looking forward to your feedback on our rebuttal. Please let us know if our responses address your concerns. We would be glad to make any further explanation and clarification.
>
> Thanks,
>
> The authors

---

> ### Author Response · Authors · 2025-08-07
>
> Dear Reviewer 25Zo,
>
> We sincerely appreciate the time and effort you have dedicated to reviewing our work. As the discussion period approaches its conclusion, we would like to kindly follow up to confirm whether our responses have adequately addressed your concerns. If there is any aspect requiring further clarification, we would be more than happy to provide additional explanations and continue the discussion.
>
> We greatly appreciate your valuable feedback and look forward to hearing from you.
>
> Best regards,
>
> The authors

---

> ### Author Response · Authors · 2025-08-08
>
> Dear Reviewer 25Zo,
>
> Thank you again for the time and thought you have devoted to reviewing our work. With less than a day remaining in the discussion period, we would like to kindly check whether our responses have sufficiently addressed your concerns. If any points would benefit from further clarification, we would be happy to provide additional explanations promptly.
>
> We sincerely appreciate your feedback and look forward to your reply.
>
> Best regards,
>
> The authors

---

### Official Review · Reviewer_gzhj · 2025-07-02

**Clarity:** 3
**Significance:** 3
**Originality:** 3
**Rating:** 5
**Confidence:** 3

**Summary:**

This paper introduces U-CAN, a novel unsupervised framework for denoising 3D point clouds without needing clean reference data. [24] The method trains a neural network using a "Noise-to-Noise" matching scheme, where the goal is to make the denoised version of one noisy point cloud resemble another noisy observation of the same object. [25, 54] Key to its success is a new loss function based on Earth Moving Distance (EMD) to establish point correspondences, and a "Denoising Consistency" constraint that ensures predictions from different noisy inputs remain consistent. The authors demonstrate that U-CAN significantly outperforms state-of-the-art unsupervised methods and achieves comparable results to supervised techniques in point cloud denoising, upsampling, and even 2D image denoising.

**Questions:**

- The choice of Earth Mover's Distance (EMD) is well-justified by its ability to find one-to-one correspondences, which is critical for the Noise-to-Noise matching loss. However, EMD is known to be computationally expensive. Can you give a rough number on the computational and training time of the denoise model.

- Can you clarify how are different noisy observations P_a and P_b generated?

- Since U-CAN performs an 1-1 matching between noisy and denoised point clouds (PC), I am curious how are the noisy points mapped to the denoised ones. Would the cluster of noisy points mapped into their closest point from the denoised PC?

**Ethical Concerns:**

["NO or VERY MINOR ethics concerns only"]

**Limitations:**

See the questions.

**Quality:**

3

**Strengths And Weaknesses:**

**Strength:**
- The work introduces U-CAN, an innovative unsupervised framework that learns to denoise point clouds without requiring pairs of noisy and clean data, which are often difficult and expensive to create.

- The core concept of denoising consistency is shown to be a general principle that can be transferred to improve unsupervised 2D image denoising as well.

- The paper is well presented with solid experiment results.

---

> ### Author Rebuttal · Authors · 2025-07-31
>
> We deeply appreciate the reviewer gzhj for the thoughtful feedback and time invested in evaluating our work. We respond to each question below.
>
> **Q1: The computational cost and training time of U-CAN.**
>
> As the reviewer correctly pointed out, the computational cost of EMD increases substantially with the number of points. We address this problem by learning point denoising at the patch level, where EMD is computed only between local point cloud patches containing a relatively small number of points (around 1,000). This strategy significantly reduces training cost and leads to more robust and efficient denoising. We conducted an ablation study to investigate the impact of varying patch sizes at Sec.D and Table 11 of the Appendix, where the results indicate that a patch size of 1,000 points is optimal for U-CAN. This configuration supports efficient training with EMD, where the convergence of U-CAN takes only about 25 hours on a single NVIDIA 3090 GPU. For reference, the training of ScoreD-TTD takes about 40 hours on a single NVIDIA 3090 GPU.
>
> For the reviewer’s convenience, we include Table 11 from the Appendix below.
>
> ***Table 11 of the Appendix: Ablation studies on point cloud patch sizes.***
>
> | Dataset: PU |          |   10K, 1%   |         |   10K, 2%   |         |   10K, 3%   |         |
> |-------------|----------|-------------|---------|-------------|---------|-------------|---------|
> | Ablation    |          |     CD      |  P2M    |     CD      |  P2M    |     CD      |  P2M    |
> | 250         |          |    2.821    | 1.302   |    3.327    | 1.515   |    4.239    | 2.152   |
> | 500         |          |    2.843    | 1.432   |    3.390    | 1.703   |    3.898    | 2.004   |
> | **1000**    |          | **2.497**   |**1.105**| **3.234**   |**1.255**| **3.666**   |**1.842**|
> | 2000        |          |    2.616    | 1.175   |    3.247    | 1.547   |    3.880    | 1.994   |
>
> Moreover, we clarify that EMD is used only during training and does not affect the inference efficiency of U-CAN. Table A below presents a comparison of inference times between U-CAN, state-of-the-art unsupervised baselines (ScoreDenoise-TTD, DMR-TTD), the supervised baseline PCN, and the traditional algorithm GLR.
>
> ***Table A. Runtime comparisons under PU-Net dataset.***
>
> | Method        | Time (s) |
> |---------------|----------|
> | GLR (Traditional)          | 101.20    |
> | PCN (Supervised)          | 167.75   |
> | DMR-TTD   | 11.08    |
> | ScoreDenoise-TTD  | 18.70     |
> | **Ours**      | **10.90**     |
>
> **Q2: How are different noisy observations generated.**
>
> The noisy observations $P_a$ and $P_b$ can be achieve by: a) multiple noisy observations of the same object. The manner is supported by modern LiDAR systems which can capture multiple (10-30) noisy observations per second. b) several subsets of a dense noisy observation. For example, we can achieve 5 training instances of 10k points from a noisy observation of 50k points. An ablation study of training U-CAN with these two input patterns can be found on Table 12 of the Appendix, where the results demonstrate the effectiveness of both kinds of input.
>
> For the reviewer’s convenience, we include Table 12 from the Appendix below.
>
> ***Table 12 of the Appendix: Ablation studies on input patterns.***
>
> | Dataset: PU         |          | 10K, 1%   |         | 10K, 2%   |         | 10K, 3%   |         |
> |---------------------|----------|----------|---------|----------|---------|----------|---------|
> | Ablation            |          | CD       | P2M     | CD       | P2M     | CD       | P2M     |
> | Multiple 𝒫          |          | **2.497** | **1.105** | **3.234** | **1.255** | 3.666 | **1.842** |
> | Subsets of One 𝒫    |          | 2.580    | 1.231   | 3.287    | 1.661   | **3.605** | 1.869   |
>
> **Q3: The 1-1 matching pattern between noisy and denoised point clouds.**
>
> Yes, your understanding is correct: clusters of noisy points $P_a$ are mapped to their "closest" points in the denoised point cloud of another noisy observation $P_b$, based on Earth Mover's Distance (EMD). It is important to note that "closest" here is defined by EMD, which seeks to minimize the total one-to-one transport cost from $P_a$ to the denoised $P_b$, rather than the point-wise Euclidean distance. Mapping based on Euclidean distance is consistent with the implementation of Chamfer Distance, which has been shown to fail to converge under the noise-to-noise matching scheme in unsupervised point cloud denoising, leading to a severe degradation of CD/P2M from 2.497/1.105 to 15.14/11.36, as demonstrated in Table 4 of the ablation study.
>
> The core insight of the noise-to-noise matching scheme is that, by aligning one noisy observation with the denoised prediction of another observation of the same shape in a one-to-one manner, the underlying clean structure can be statistically inferred.
> However, we also observe that this strategy may result in inconsistencies across different denoising results. To address this, we propose a consistency-aware constraint that minimizes geometric differences between pairs of denoised predictions, leading to more robust denoising performance.

---

> > ### Comment · Reviewer_gzhj · 2025-08-09
> >
> > Thank the authors for addressing my questions, I am good to keep my score as accept!

---

> ### Author Response · Authors · 2025-08-09
>
> Dear Reviewer gzhj,
>
> Many thanks for all the invaluable feedback and positive assessment! We really appreciate your expertise and time invested in evaluating our work.
>
> Best regards,
>
> The authors

---

### Author Response · Authors · 2025-08-04
**We will be happy to take any questions**

Dear reviewers,

We sincerely appreciate your comments and expertise. Please let us know if there is anything we can clarify further. We would be happy to take this opportunity to discuss with you.

Thanks,

The authors

---

### Note · Authors · 2025-08-15

Dear NeurIPS 2025 Area Chairs and Reviewers,

Thank you for your time and effort invested in evaluating our work. We sincerely appreciate all the valuable feedback and constructive comments provided during the review process, as well as the engaging rebuttal and discussion sessions. We are pleased to have addressed most of the reviewers’ concerns, and it is encouraging to hear that all reviewers participating in the discussion (Reviewer gzhj, Reviewer y6U3, and Reviewer hTqa) are inclined to recommend our paper for acceptance. While we did not receive further comments from Reviewer 25Zo, we believe that our comprehensive responses and additional experiments have sufficiently addressed the Reviewer 25Zo's questions.

We conducted comprehensive experiments to address the reviewers’ concerns.

- **Table A** presents runtime comparisons on the PU-Net dataset, demonstrating the efficiency of our method and addressing the questions on computational cost raised by Reviewers gzhj and 25Zo.
- **Table B** provides results on the indoor ScanNet dataset, showing U-CAN’s superior performance on real-world scanned scene point clouds and addressing the questions on real-world scenarios from Reviewers 25Zo and hTqa.
- **Table C** reports comparisons on the real-world scanned garment dataset DeepFashion3D, addressing Reviewer y6U3’s question on representation differences with NoiseMap.
- **Table D** presents an ablation study of the multi-step framework design, addressing Reviewer y6U3’s question regarding loss function settings.
- **Table E** updates the P2M metric comparison, addressing Reviewer y6U3’s question on evaluation under low noise levels.
- **Table F** shows experiments where the U-CAN backbone is replaced with PointNet++ and PointTransformer, demonstrating robustness across different backbones and addressing Reviewer hTqa’s question on framework adaptability.

We are grateful that our additional experiments and detailed responses have resolved the concerns of the Reviewers. The constructive feedback has significantly strengthened our paper. We believe that U-CAN provides a promising solution for consistent, efficient, and robust unsupervised point cloud denoising. We will incorporate the experiments, visualizations, and implementation details discussed in the rebuttal into the final version.

Best regards,

The authors

---

### Decision · Program_Chairs · 2025-09-17

**Decision:**

Accept (poster)

**Comment:**

The paper initially received mixed reviews. The major concerns were:

1) computational cost of EMD? [gzhj]
2) needs more description about the noise distributions [25Zo]
3) performance under real world noise? [25Zo]
4) does U-can have advantage in model complexity? [25Zo]
5) core ideas from Noise2Noise for 2D image denoising and multi-step denoising [25Zo, y6U3]
6) missing baseline comparison [y6U3]
7) Why does the proposed method perform worse in P2M under low-noise levels? [y6U3]
8) missing ablation study on backbone networks [hTqa]
9) missing evaluation on indoor scenes [hTqa]

The authors wrote a response to address these points. After the discussion, gzhj and hTqa were satisfied and kept their ratings. y6U3 was also satisfied and updated to border accept. However, y6U3 still had reservations that the underlying principle is from Noise2NoiseMapping for 2d images, but acknowledged the differences in learning scheme, framework, and novelty of enforcing cross-scan consistency. 25Zo maintained their original rating of borderline reject, but did not give any particular comments about how the rebuttal failed to address their concerns. The final ratings were 4534.

After reading the paper, reviews, rebuttal, and discussions, the AC thought that the paper was sufficiently novel as it adapts the Noise2NoiseMapping principle from 2d images to point clouds, and addresses unique challenges that arise from this. The experiment results are compelling. The AC thought that 25Zo's comments were addressed well enough. Overall, the AC agrees with the positive reviewers and recommends Accept.

The authors should revise the paper according to the reviews, rebuttal, and discussion.